# MAV Localization in Large-Scale Environments: A Decoupled Optimization/Filtering Approach

**DOI:** 10.3390/s23010516

**Published:** 2023-01-03

**Authors:** Abanob Soliman, Hicham Hadj-Abdelkader, Fabien Bonardi, Samia Bouchafa, Désiré Sidibé

**Affiliations:** IBISC Laboratory, Université d’Evry-Paris Saclay, 91020 Evry-Courcouronnes, France

**Keywords:** MAV, multimodal sensing, localization, odometry, visual drifts, sensor fusion, Kalman filter, calibration, optimization

## Abstract

Developing new sensor fusion algorithms has become indispensable to tackle the daunting problem of GPS-aided micro aerial vehicle (MAV) localization in large-scale landscapes. Sensor fusion should guarantee high-accuracy estimation with the least amount of system delay. Towards this goal, we propose a linear optimal state estimation approach for the MAV to avoid complicated and high-latency calculations and an immediate metric-scale recovery paradigm that uses low-rate noisy GPS measurements when available. Our proposed strategy shows how the vision sensor can quickly bootstrap a pose that has been arbitrarily scaled and recovered from various drifts that affect vision-based algorithms. We can consider the camera as a “black-box” pose estimator thanks to our proposed optimization/filtering-based methodology. This maintains the sensor fusion algorithm’s computational complexity and makes it suitable for MAV’s long-term operations in expansive areas. Due to the limited global tracking and localization data from the GPS sensors, our proposal on MAV’s localization solution considers the sensor measurement uncertainty constraints under such circumstances. Extensive quantitative and qualitative analyses utilizing real-world and large-scale MAV sequences demonstrate the higher performance of our technique in comparison to most recent state-of-the-art algorithms in terms of trajectory estimation accuracy and system latency.

## 1. Introduction

Robust localization of micro aerial vehicles (MAVs) in uncharted large-scale areas can rely on complementary data gathered by many sensor modalities. The study of simultaneous localization and mapping (SLAM), primarily used for MAV navigation in expansive and dynamic settings, may be enriched and expanded by using multi-modal datasets [1]. These settings have certain traits, such as the dynamic range of the scene’s object intensities. For instance, mapping a small interior space with adequate illumination might be of more outstanding quality than mapping a rural area at night with heavy rain, wind, and fog (outdoors dynamic environment). The benefits of multimodal approaches become apparent when systems rely on sensors with a high dynamic range and strong sensing capabilities, such as event cameras, LiDARs, or radars, or typical inexpensive cameras fused with other sensor modalities such as the inertial measurement units (IMUs) and GPS sensors. These multimodal approaches can indeed fill some lack of data during scene mapping and MAV localization.

Towards this aim, we develop a trustworthy (quick and precise) localization solution that utilizes information from three sensor modalities: camera frame data, IMU measurements, and GPS readings. Nevertheless, the GPS sensor readings are consistently slower and noisier than those from the IMU or camera modules, and they frequently experience signal loss in GPS-restricted locations. Therefore, a localization system that depends on GPS data must perform effectively when GPS readings are lost.

Visual-inertial odometry (VIO) is one of the most mature and well-established approaches in the localization field [2,3,4]. Efficient visual odometry can be achieved using a high-quality perception of the surroundings. Sensors performing this perception task can differ in their nature of data collection. On the one hand, the most common visual odometry sensors are cameras such as RGB cameras [5], event cameras [6], and RGB-D cameras [7]. On the other hand, using the LiDAR sensor [8] can provide point clouds and a GPS sensor [9,10] can locate the MAV using satellite signals triangulation, as represented in Figure 1.

The accuracy of the state estimation process relies on an error-state extended Kalman filter (ES-EKF) and the bootstrapping quality of its states. A well-established IMU-based state estimator initialization technique was discussed in [5]. In this bootstrapping method, the global metric scale of the trajectory and the IMU-camera gravity alignment is optimized using a specific amount of IMU readings preintegration combined with an initial up-to-scale trajectory estimated using the camera only. This bootstrapping process is prone to failure due to insufficient IMU excitation, especially when the MAV navigates in a planar terrain.

The MAV should contain a localization system that continually calculates the pose with high accuracy and low latency during search and rescue missions, for instance. The MAV is equipped with restricted resources regarding the data processing unit and the limited power source capacity for long-term navigation operations in large-scale situations. In light of this, the state estimate approach should consistently have low computational complexity and resist sensor readings that deviate from the norm.

Our work’s main contribution to tackle the aforementioned challenges is three-fold:-In the case of state estimator initialization failure, we propose a unique instant bootstrapping technique based on continuous-time manifold optimization via pose graph optimization (PGO) and range factors, which depends on low-rate GPS signals.-A closed-form estimation method without nonlinear optimization during IMU/CAM fusion produces a reduced system latency with constant CPU computing complexity. The mathematical modeling of a linear ES-EKF with a precise and quick gyroscope integration strategy accounts for the simplicity of our proposed localization solution.-The EuRoC benchmark [12], for MAV localization assessment in indoor environments, and the Fast Flight dataset [11], for large-scale outdoor environments, are two real-world publicly available benchmarks on which our IMU/GPS-CAM fusion system has been thoroughly tested. With thorough ablation investigations into the role of each sensor modality in the overall accuracy of the state estimation process, the assessment is conducted using the most recent state-of-the-art visual-inertial odometry methodologies.

## 2. Related Work

### 2.1. Sensor Fusion

Figure 2 presents a global overview of the current state-of-the-art approaches for localization. The ability to continually estimate the robot’s ego-motion (position and orientation) over time is a significant difficulty in autonomous navigation, path planning, object tracking, and collision avoidance platforms [13]. The Global Positioning System (GPS) is a well-known localization method applied to several autonomous system domains. One kind of global navigation satellite system (GNSS) is GPS [10]. GPS is used as a self-localization source, such as for MAVs security applications, and gives any user with a GPS receiver positional information with meter-level precision. The satellite signal blockage, high noise levels, multipath effects, and other issues with GPS, on the other hand, make it a less trustworthy alternative sensor for self-localization modules. However, real-time kinematic (RTK) and precise point positioning (PPP) [9], two GPS technologies that are rapidly developing, can provide locations with decimeter- or centimeter-level precision.

The effectiveness of GPS satellite signals heavily depends on the surrounding environment; it works best in locations with clear skies and is ineffective for inside navigation since walls and other obstacles impede it [14]. This makes the GPS module an unsuitable primary sensor for reliable autonomous vehicle localization under adverse weather and environmental conditions. Hence, the fusion of GPS signals with other inertial and/or visual sensors is indispensable for a reliable localization solution, especially in such environments. The state-of-the-art sensor fusion systems are differentiated into two prominent families: loosely- [15], and tightly-coupled [16] fusion strategies. In loosely-coupled fusion, the camera frames for pose estimation are processed as a black-box. A filter or an optimization model is developed to fuse the arbitrary-scaled poses from the visual sensor with the noisy metric-scaled re-integrated IMU readings [17].

On the contrary, in the tightly coupled approach, scene information from the visual sensor is fused with the IMU measurements (linear accelerations and angular velocities) using a fusion filter or an optimization model that estimates the metric-scaled pose, visual odometry scale factor, IMU biases, and visual drift between the IMU-camera inertial frames. One of the prominent advantages of a tightly-coupled fusion scheme is that it can estimate accurate scene information to reconstruct a precise scene map, along with providing the SLAM system with high confidence in loop closure during re-localization situations.

### 2.2. Fusion Strategies

The two sensor fusion strategies (loosely and tightly coupled) have two main execution techniques: filter-based and optimization-based execution. Some filter-based state-of-the-art approaches are deterministic, such as MSCKF [18], S-MSCKF [11], S-UKF-LG/S-IEKF [19], and ROVIO [20]. At the same time, alternative strategies can be based on nondeterministic filters such as particle filters [21].

Optimization-based methods such as VINS-Mono [22], OKVIS [23], ORB-SLAM [24], and BASALT [25], can be deterministic or non-deterministic based on the optimization strategy and the convergence constraints. The estimation and robustness of visual localization frameworks have advanced significantly in recent decades, and this development may be furthered by tightly integrating visual and inertial data. Most methods integrate data utilizing optimization methods or filtering-based procedures.

Filtering approaches are ideally suited to real-time applications [26,27], which is the main emphasis of this study. In contrast, optimization-based methods are more precise but often have a more extensive processing complexity. The observability-constrained technique addresses the consistency issue, a shortcoming of traditional VIO filter-based algorithms [28]. The EKF/MSCKF and its cutting-edge variations are among the most widely used solutions because they effectively balance accuracy and computational complexity.

A recent study [29] shows that if the air mass’s random character is considered, the EKF system states of an MAV are observable. The drag and lift forces on the MAV will directly impact the projected pose and velocity due to the nature of air mass randomization. To make an online update for the uncertainties brought on by these random effects on the precise position of the sensors’ reference frames, we contribute with a visual drift augmentation technique to our EKF measurement model. The EKF’s ability to tolerate significant disturbances in the MAV’s velocity state variable and still converge to the undisturbed estimates is what we target.

### 2.3. Visual Odometry

The main objective of a visual odometry solution is to perform an accurate and precise localization of the robot (ground or aerial vehicle) to estimate its pose during the navigation task. Estimated poses can be on either discrete- or continuous-time manifolds. Cioffi et al. [30] studied the reliability of the estimated poses on both manifolds using IMU/Visual/GPS sensors. They came to an important conclusion: similar results are produced by the two representations when the camera and IMU are time-synchronized.

In [13], the sliding window pose-graph optimization of the most recent robot states uses global position data with poses predicted by a VIO method. Like [15], pose-graph optimization employs an independent VIO technique to generate pose estimations fused with GPS data. In contrast to [13], the pose-graph in [15] includes an extra node representing the local coordinate frame’s origin to confine the absolute orientation. However, these methods are loosely connected, meaning that a separate VIO algorithm generates the relative pose estimations. Inspired by [13,15], we present a loosely coupled strategy that considers the correlations between all measures by including them in a hybrid optimization and filtering problem.

It is demonstrated in [23] that considering all measurement correlations is essential for high-precision estimations in the visual-inertial situation. A tightly coupled sliding window optimization for visual and inertial data with loosely connected GPS refinement is presented in [14]. The GPS readings are given the same timestamp as the temporally nearest image to be included in the sliding window because it is believed that they would only be accessible at low rates. As opposed to [14], we efficiently compute the global positional factors by closely coupling the global position measurements using the Runge–Kutta 4th-order gyroscope preintegration scheme [31]. This enables the sliding window to incorporate numerous global parameters and each keyframe with barely any additional processing load.

### 2.4. Methodology Background

We highlight the methodology that inspires our study in blue-dashed rectangles in Figure 2. Where the loosely coupled fusion strategy [32] is adopted to keep constant computational complexity for real-time performance, along with adding a reset mode for the framework, as discussed in [33] as well as an online IMU-camera extrinsic calibration paradigm [4]. Integrating the IMU/GPS readings with the global shutter visual sensor monocular frames raises our localization solution’s accuracy level, leveraging the MAV’s inertial and global localization information.

Pushing the limits of the extended Kalman filter to raise the robustness of our localization solution towards a resilient system, we leverage the high accuracy of the optimization to initialize the filter pose states using a novel instant approach utilizing the low-rate noisy GPS readings when available. Sensor fusion on continuous-time (CT) manifolds, such as B-splines [34], suffers from a high execution complexity, especially with the time derivatives of high-order manifolds for integrating the IMU measurements in the estimation process. Hence, in our novel method, we avoid this dilemma with a simple spline-fitting approach for the GPS readings during the data pre-processing stage.

## 3. System Architecture

Our core sensor setup consists of an inertial navigation sensor (IMU), a global positioning sensor (GPS), and a monocular camera, as illustrated in Figure 3. The pipeline starts with the data acquisition and pre-processing for the initialization process, as discussed in Section 3.1. The initialization is an optimization-based phase (see Algorithm 1) with a considerably low complexity and processing time whose output is an instant metric-scaled pose estimated from the camera, GPS, and gyroscope readings. Then, an ES-EKF (see Algorithm 2) whose dynamic model is given in Section 3.2, is applied to estimate all the system states, including the MAV’s trajectory, velocity, and a scale factor to recover the initially estimated trajectory in the case of GPS readings loss. Finally, we present the measurement model in Section 3.3 with a novel false pose augmentation paradigm to ensure the observability of all the filter states, as analyzed in Appendix A.
**Algorithm 1** Bootstrapping: Pose Graph Optimization and Range Factors **Input:** RGB frames (*c*), camera matrix ($K_{c}$), GPS readings (DT-GPS), IMU readings (I) **Output:** Metric-scaled trajectory ($T_{vc}\left[p_{v}^{c},\;q_{v}^{c}\right]\in SE\left(3\right)$)1:$T_{vc}^{0}⇐\mathrm{KLT}\text{-}\mathrm{VO}\;(c,\;K_{c})$                 ▹ Arbitrary-scaled pose2:p(u)⇐spline_fit (DT-GPS)               ▹ CT-GPS by Equation (Equation 4)3:$\left[\phi ,\theta ,\psi ]⇐\mathrm{RK}4\;(I_{gyro}\left(\omega \right)\right)$                 ▹ Initial orientations4:**while***not**converged***do**                 ▹ Initial trajectory optimization5:    $T_{vc}⇐\mathrm{optimize}\;(T_{vc}^{0},\;p(u),\;\left[\phi ,\theta ,\psi \right])$               ▹ Equation (Equation 6)6:**end while**

The state representation is a 31-element state vector X:(1)$$X={\left[{p_{w}^{i}}^{⊤}\;{v_{w}^{i}}^{⊤}\;{q_{w}^{i}}^{⊤}\;{b_{\omega }}^{⊤}\;{b_{a}}^{⊤}\;\lambda \;{p_{i}^{c}}^{⊤}\;{q_{i}^{c}}^{⊤}\;{p_{v}^{w}}^{⊤}\;{q_{v}^{w}}^{⊤}\right]}^{⊤},$$
where $p_{w}^{i}$ is the position of the IMU in the world frame (world frame is a gravity-aligned frame.) (w), its velocity $v_{w}^{i}$, and its attitude rotation quaternion $q_{w}^{i}$ describing a rotation from the IMU frame (i) into the world frame (w). $b_{\omega }$ and $b_{a}$ are the gyro and acceleration biases along with the visual odometry scale factor λ. $R_{\left(q\right)}$ is the quaternion *q* rotational matrix, *g* is the gravity vector aligned with the world frame (w), and Ω(ω) is the quaternion-multiplication matrix of ω.

The IMU/camera calibration states are the rotation from the camera frame into the IMU frame $q_{i}^{c}$, and the position of the camera center with regard to the IMU frame $p_{i}^{c}$.

Finally, the visual attitude drifts between the black-boxed visual frame (vision frame is the frame to which the camera pose is estimated in the black-box vision framework) (v) and the world inertial frame (w) are reflected in $q_{v}^{w}$ and the translational ones in $p_{v}^{w}$. We assume that all the visual drifts are spatial without any temporal drifts, i.e., the IMU and the camera have synchronized timestamps.
**Algorithm 2** End-to-End State Estimation Scheme     **Input:** IMU readings, initial optimized trajectory $T_{vc}$     **Output:** FilterStates $X=\left\{\lambda ,K_{i}\left[b_{a},\;b_{\omega }\right],T_{ic},T_{wv},T_{wi},v_{wi}\right\},\;\forall T\left[p,q\right]\in SE\left(3\right)$1:$P,Q_{c},R$_initialization, FilterStates_initialization2:ErrorStates_initialization=03:**while** imuRead **do**4:    Read LastStep (k) P, FilterStates, ErrorStates5:    Read LastStep (k) IMU (Accel, Gyro) values6:    Read current (k+1) IMU (Accel, Gyro) values7:    Step 1: Propagate IMU states                    ▹ Equation (Equation 12)8:    Step 2: Calculate $F_{d}$ and $Q_{d}$               ▹ Equations (Equation 14) and (Equation 16)9:    Step 3: Compute P state covariance matrix              ▹ Equation (Equation 18)10:   **if** camRead **then**11:        Read current (k+1) CAM $T_{vc}$ values               ▹ Metric-scaled pose12:        Step 4: Estimate false pose                     ▹ Equation (Equation 19)13:        Step 5: Calculate $\tilde{z}$, H                       ▹ Equation (Equation 20)14:        Step 6: Calculate S, K, ErrorStates $\hat{\tilde{x}}$, P            ▹ Equations (Equation 26) and (Equation 27)15:        Step 7: Update: FilterStates += ErrorStates16:        Step 8: RESET $\hat{\tilde{x}}=0$, P                      ▹ Equation (Equation 29)17:    **end if**18:**end while**

The corresponding 28-elements error state vector is defined by:(2)$$\tilde{x}={\left[{\Delta p_{w}^{i}}^{⊤}\;{\Delta v_{w}^{i}}^{⊤}\;{\delta \theta_{w}^{i}}^{⊤}\;{\Delta b_{\omega }}^{⊤}\;{\Delta b_{a}}^{⊤}\;\Delta \lambda \;{\Delta p_{i}^{c}}^{⊤}\;{\delta \theta_{i}^{c}}^{⊤}\;{\Delta p_{v}^{w}}^{⊤}\;{\delta \theta_{v}^{w}}^{⊤}\right]}^{⊤},$$
as the difference of an estimate $\hat{x}$ to its quantity *x*, i.e., $\tilde{x}=x-\hat{x}$. We apply this to all state variables except the error quaternions, which are defined by:(3)$$\delta q_{y}^{x}=q_{y}^{x}⊗{\hat{q}}_{y}^{x}\approx {\left[\begin{array}{cc} \frac{1}{2}\delta \theta_{y}^{x} & 1 \end{array}\right]}^{⊤}.$$

This error quaternion representation increases the numerical stability of the estimation process and handles the quaternion in its minimal representation [35].

### 3.1. State Estimator Initialization

An incremental structure from motion (SfM) algorithm [36] is applied to the acquired image frames, whose goal is to retrieve the camera poses and the 3D structure of the scene, based on the five-point algorithm proposed in [37]. ORB features are detected, and the highest quality points are tracked between 10 consecutive frames using the KLT method [38].

To solve the arbitrary-scale problem of the camera trajectory only, we apply an on-manifold cumulative B-spline (https://github.com/AbanobSoliman/B-splines (accessed on 1 October 2022)) interpolation [34] to synthesize a very smooth continuous-time (CT) trajectory in $R^{3}$ from the low-rate noisy GPS readings.

The matrix form for the cumulative B-spline manifold of order k=n+1, where *n* is the spline degree, is modeled at $t\in [t_{i},t_{i+k-1}]$ as:(4)$$p\left(u\right)=p_{i}+\sum_{j=1}^{k-1}{\tilde{B}}_{j}^{\left(k\right)}.{\bar{u}}_{j}^{\left(k\right)}.d_{j}^{i},$$
where $p\left(u\right)\in R^{3}$ is the continuous-time B-spline increment that interpolates *k* GPS measurements on the normalized unit of time $u\left(t\right):=\left(t-t_{i}\right)/\Delta t_{s}-P_{n}$ with $1/\Delta t_{s}$ denoting the spline generation frequency and $P_{n}$ being the pose number that contributes to the current spline segment $P_{n}\in \left[0,\cdots ,k-1\right]$. $p_{i}$ is the initial discrete-time (DT) GPS location measurement at time $t_{i}$. The term $d_{j}^{i}=p_{i+j}-p_{i+j-1}$ is the difference vector between two consecutive DT-GPS readings. The matrix ${\tilde{B}}_{j}^{\left(k\right)}$ is the cumulative basis blending and ${\bar{u}}_{j}^{\left(k\right)}$ is the normalized time vector, both of which are defined as:(5)$$\begin{array}{c} {\tilde{B}}_{j}^{\left(k\right)}={\tilde{b}}_{j,n}^{\left(k\right)}=\sum_{s=j}^{k-1}b_{s,n}^{\left(k\right)},\;\\ b_{s,n}^{\left(k\right)}=\frac{C_{k-1}^{n}}{(k-1)!}\sum_{l=s}^{k-1}{\left(-1\right)}^{l-s}C_{k}^{l-s}{\left(k-1-l\right)}^{k-1-n},\;\\ {\bar{u}}_{j}^{\left(k\right)}={\left[u^{0},\cdots ,\;u^{k-1},\;u^{k}\right]}^{⊤},\;u\in \left[0,\cdots ,1\right]. \end{array}$$

Our GPS-IMU aided initialization system comprises two optimization factors: the first is a pose graph optimization (PGO) factor $r^{p}$ that optimizes the 6-DoF of every pose, whereas the second is a range factor $r^{s}$ that constraints the translation limits between every two KLT-VO poses. Hence, the metric scale of the visual odometry pose is recovered using the gyroscope and GPS readings, leveraging the high accuracy of the optimization process. An illustrative scheme for the initialization process factor graph is shown in Figure 4.

Level 1’s objective function $L^{p,s}$ is modeled as:(6)$$L^{p,s}=\underset{T_{wi}}{arg min}\left[\sum_{\left(i,j\right)}^{N}\left({\left|\left|r^{p}\left(i,j\right)\right|\right|}_{\Sigma_{i,j}^{p}}^{2}+{\left|\left|r^{s}\left(i,j\right)\right|\right|}_{\Sigma_{i,j}^{s}}^{2}\right)\right].$$

$\Sigma_{i,j}^{p},\Sigma_{i,j}^{s}$ are the information matrices associated with the GPS readings covariance, reflecting the PGO and Range factors noises on the global metric scale estimation process between two RGB-D aligned frames.

**Pose Graph Optimization (PGO) factor.** The PGO is a 6-DoF factor that controls the relative pose error between two consecutive edges i,j and is formulated as:(7)$$r^{p}={\left|\left|\left({\hat{T}}_{i}^{-1}{\hat{T}}_{j}\right)⊖\Delta T_{ij}^{\omega ,GPS}\right|\right|}_{2}$$
where ${\left|\left|.\right|\right|}_{2}$ is the L2-norm, ${\hat{T}}_{i,j}\in SE\left(3\right)$ is the $T_{wi}^{0}$ estimated from the front-end pipeline at frames i,j. The operator ⊖ is the SE(3) logarithmic map as defined in [39]. The error transformation $\Delta T_{ij}^{\omega ,GPS}\left[\delta R_{ij}^{\omega },\delta p_{ij}^{GPS}\right]\in \mathrm{se}\left(3\right)$, where $\delta p_{ij}^{GPS}=p_{j}-p_{i}$ is the CT-GPS measurement increment and $\delta R_{ij}^{\omega }={\left[\delta \phi ,\delta \theta ,\delta \psi \right]}^{⊤}\in \mathrm{so}\left(3\right)$ is the gyroscope integrated increment $\delta R_{ij}^{\omega }=\int_{k=i}^{j}(\omega_{k}).dk$ using the Runge–Kutta 4th order (RK4) integration method [31] between the keyframes *i* and *j*.

**Range factor.** The range factor limits the front-end visual drift and keeps the global metric scale under control within a sensible range defined by the GPS signal and is formulated as:(8)$$r^{s}={\left|\left|{\left|\left|{\hat{t}}_{j}-{\hat{t}}_{i}\right|\right|}_{2}-{\left|\left|p_{j}^{GPS}-p_{i}^{GPS}\right|\right|}_{2}\right|\right|}_{2}$$
where ${\hat{t}}_{i,j},p_{i,j}^{GPS}\in R^{3}$ are the translation vectors of two consecutive front-end (KLT-VO) poses and CT-GPS signals, respectively.

### 3.2. Dynamic Model

The core state estimation is performed by fusing the RGB camera frames and the IMU reading using an error states extended Kalman filter (ES-EKF). Figure 5 illustrates the inter-sensor extrinsic relation between the IMU/GPS sensors and a monocular camera.

To use the linear states estimator, we assume that the IMU measurements contain a particular bias $b_{a}\in N\left(0,\sigma_{ba}\right)$, $b_{\omega }\in N\left(0,\sigma_{b\omega }\right)$ and a white Gaussian noise $n_{a}\in N\left(0,\sigma_{a}\right)$, $n_{\omega }\in N\left(0,\sigma_{\omega }\right)$.

Thus, the real angular velocities ω and accelerations *a* in the IMU body frame (i) can be written as:(9)$$\begin{array}{ccc} \omega =\omega_{m}-b_{\omega }-n_{\omega } & and & a=a_{m}-b_{a}-n_{a}, \end{array}$$
where the subscript *m* denotes the measured value. The dynamics of the non-static biases are modeled as a random process:(10)$$\begin{array}{cc} \dot{b_{\omega }}=n_{b_{\omega }}, & \dot{b_{a}}=n_{b_{a}}. \end{array}$$

The standard deviation $\sigma_{b_{\omega }},\sigma_{b_{a}},\sigma_{w},\sigma_{a}$ values are generally given by the IMU manufacturer’s data in Allan deviation plots. For discrete time steps, it will be applied in the filter. We need to convert these values according to their units:(11)$$\begin{array}{cc} d\sigma_{\omega ,a}^{2}=\frac{\sigma_{\omega ,a}^{2}}{\nabla t}, & d\sigma_{b_{\omega ,a}}^{2}=\sigma_{b_{\omega ,a}}^{2}*\nabla t. \end{array}$$

The following differential equations govern IMU state propagation:(12)$$\begin{array}{c} \dot{p_{w}^{i}}=v_{w}^{i},\;\\ \dot{v_{w}^{i}}=R_{\left(q_{w}^{i}\right)}^{⊤}\left(a_{m}-b_{a}-n_{a}\right)-g,\;\\ \dot{q_{w}^{i}}=\frac{1}{2}\Omega \left(\omega_{m}-b_{\omega }-n_{\omega }\right)q_{w}^{i},\;\\ \dot{b_{\omega }}=n_{b_{\omega }},\;\dot{b_{a}}=n_{b_{a}},\;\dot{\lambda }=0,\;\\ \dot{p_{i}^{c}}=0,\;\dot{q_{i}^{c}}=0,\;\dot{p_{v}^{w}}=0,\;\dot{q_{v}^{w}}=0, \end{array}$$

For the quaternion integration inside the ES-EKF, we use the first-order integrator defined in [35] as:(13)$$\begin{array}{c} \bar{w}=\frac{\omega_{k+1}+\omega_{k}}{2},\;\kappa =\frac{1}{2}.\Omega \left(\bar{\omega }\right).\Delta t,\;\\ {\hat{q_{w}^{i}}}_{k+1}=\left[e^{\kappa }+\frac{\Delta t^{2}}{48}\left(\Omega \left(\omega_{k+1}\right).\Omega \left(\omega_{k}\right)-\Omega \left(\omega_{k}\right).\Omega \left(\omega_{k+1}\right)\right)\right].{\hat{q_{w}^{i}}}_{k}. \end{array}$$
where the hat term ^ means the estimated value. The exponential term $e^{\kappa }$ is expanded by the Maclaurin series.

The states transition matrix $F_{d}$ is modeled as:(14)$$F_{d}=\left[\begin{array}{cccccc} I_{d_{3}} & \Delta t & A & B & -R_{\left(\hat{q_{w}^{i}}\right)}^{⊤}\frac{\Delta t^{2}}{2} & 0_{3\times 13}\\ 0_{3} & I_{d_{3}} & C & D & -R_{\left(\hat{q_{w}^{i}}\right)}^{⊤}\Delta t & 0_{3\times 13}\\ 0_{3} & 0_{3} & E & F & 0_{3} & 0_{3\times 13}\\ 0_{3} & 0_{3} & 0_{3} & I_{d_{3}} & 0_{3} & 0_{3\times 13}\\ 0_{3} & 0_{3} & 0_{3} & 0_{3} & I_{d_{3}} & 0_{3\times 13}\\ 0_{13\times 3} & 0_{13\times 3} & 0_{13\times 3} & 0_{13\times 3} & 0_{13\times 3} & I_{d_{13}} \end{array}\right].$$ Then, we apply the small-angle approximation for which |ω|→0 apply the de l’Hopital rule and obtain a compact solution for the six matrix blocks A,B,C,D,E,F [35]:(15)$$\begin{array}{c} A=-R_{\left(\hat{q_{w}^{i}}\right)}^{⊤}{\left\lfloor \hat{a}\right\rfloor }_{\times }\left(\frac{\Delta t^{2}}{2!}-\frac{\Delta t^{3}}{3!}{\left\lfloor \hat{\omega }\right\rfloor }_{\times }+\frac{\Delta t^{4}}{4!}{\left\lfloor \hat{\omega }\right\rfloor }_{\times }^{2}\right),\;\\ B=-R_{\left(\hat{q_{w}^{i}}\right)}^{⊤}{\left\lfloor \hat{a}\right\rfloor }_{\times }\left(\frac{-\Delta t^{3}}{3!}+\frac{\Delta t^{4}}{4!}{\left\lfloor \hat{\omega }\right\rfloor }_{\times }-\frac{\Delta t^{5}}{5!}{\left\lfloor \hat{\omega }\right\rfloor }_{\times }^{2}\right),\;\\ C=-R_{\left(\hat{q_{w}^{i}}\right)}^{⊤}{\left\lfloor \hat{a}\right\rfloor }_{\times }\left(\Delta t-\frac{\Delta t^{2}}{2!}{\left\lfloor \hat{\omega }\right\rfloor }_{\times }+\frac{\Delta t^{3}}{3!}{\left\lfloor \hat{\omega }\right\rfloor }_{\times }^{2}\right),\;\\ D=-A,\;\\ E=I_{d_{3}}-\Delta t{\left\lfloor \hat{\omega }\right\rfloor }_{\times }+\frac{\Delta t^{2}}{2!}{\left\lfloor \hat{\omega }\right\rfloor }_{\times }^{2},\;\\ F=-\Delta t+\frac{\Delta t^{2}}{2!}{\left\lfloor \hat{\omega }\right\rfloor }_{\times }-\frac{\Delta t^{3}}{3!}{\left\lfloor \hat{\omega }\right\rfloor }_{\times }^{2}, \end{array}$$
with $\hat{\omega }=\omega_{m}-\hat{b_{\omega }}$, $\hat{a}=a_{m}-\hat{b_{a}}$ and ${\left\lfloor \hat{\omega }\right\rfloor }_{\times }$, ${\left\lfloor \hat{a}\right\rfloor }_{\times }$ the skew-symmetric matrices for IMU readings.

We can now derive the discrete-time input noise covariance matrix $Q_{d}$ as: (16)$$Q_{d}=\underset{\Delta t}{ʃ}F_{d}\left(\tau \right)G_{c}Q_{c}G_{c}^{⊤}F_{d}{\left(\tau \right)}^{⊤}d\tau ,$$
where $Q_{c}$ is the CT process noise covariance, and $G_{c}$ is calculated in the form:(17)$$G_{c}=\left[\begin{array}{cccc} 0_{3} & 0_{3} & 0_{3} & 0_{3}\\ -R_{\left(\hat{q_{w}^{i}}\right)}^{⊤} & 0_{3} & 0_{3} & 0_{3}\\ 0_{3} & 0_{3} & I_{d_{3}} & 0_{3}\\ 0_{3} & 0_{3} & 0_{3} & I_{d_{3}}\\ 0_{3} & -I_{d_{3}} & 0_{3} & 0_{3}\\ 0_{13\times 3} & 0_{13\times 3} & 0_{13\times 3} & 0_{13\times 3} \end{array}\right].$$

The closed-form solution of the complete derivation of the $Q_{d}$ covariance matrix is given in detail in Appendix B.

Finally, the propagated state covariance matrix computation is defined as:(18)$$P_{k+1|k}=F_{d}P_{k|k}F_{d}^{⊤}+Q_{d}.$$

### 3.3. Measurement Model

The main contribution of our measurement model for an observable ES-EKF is the false relative pose augmentation methodology of the visual drift quaternion state at the previous time step (k) updated with the current camera measurement at a time (k+1) and modeled as:(19)$$q_{v}^{w}\left(k\right)={\hat{q}}_{w}^{i}{\left(k\right)}^{-1}⊗{\hat{q}}_{i}^{c}{\left(k\right)}^{-1}⊗q_{v}^{c}\left(k+1\right).$$

The camera position measurement model yields the position of the camera with respect to the vision frame $p_{v}^{c}$. The error in measurement modeled as ${\tilde{z}}_{p}$ and linearized as ${\tilde{z}}_{pL}$:(20)$${\tilde{z}}_{p}=z_{p}-{\hat{z}}_{p}=p_{v}^{c}-R_{\left({\hat{q}}_{v}^{w}\right)}^{⊤}\left({\hat{p}}_{w}^{i}+R_{\left({\hat{q}}_{w}^{i}\right)}^{⊤}{\hat{p}}_{i}^{c}\right)\hat{\lambda }\;\dot{=}\;{\tilde{z}}_{pL}=H_{p}\tilde{x},$$
with
(21)$$H_{p}^{⊤}=\left[\begin{array}{c} R_{\left({\hat{q}}_{v}^{w}\right)}^{⊤}\hat{\lambda }\\ 0_{3\times 3}\\ -R_{\left({\hat{q}}_{v}^{w}\right)}^{⊤}R_{\left({\hat{q}}_{w}^{i}\right)}^{⊤}{\left\lfloor {\hat{p}}_{i}^{c}\right\rfloor }_{\times }\hat{\lambda }\\ 0_{6x3}\\ R_{\left({\hat{q}}_{v}^{w}\right)}^{⊤}R_{\left({\hat{q}}_{w}^{i}\right)}^{⊤}{\hat{p}}_{i}^{c}+R_{\left({\hat{q}}_{v}^{w}\right)}^{⊤}{\hat{p}}_{w}^{i}\\ R_{\left({\hat{q}}_{v}^{w}\right)}^{⊤}R_{\left({\hat{q}}_{w}^{i}\right)}^{⊤}\hat{\lambda }\\ 0_{6x3}\\ -R_{\left({\hat{q}}_{v}^{w}\right)}^{⊤}{\left\lfloor \left({\hat{p}}_{w}^{i}+R_{\left({\hat{q}}_{w}^{i}\right)}^{⊤}{\hat{p}}_{i}^{c}\right)\hat{\lambda }\right\rfloor }_{\times } \end{array}\right],$$
using the definition of the error-quaternion
(22)$$\begin{array}{c} q_{w}^{i}=\delta q_{w}^{i}⊗{\hat{q}}_{w}^{i},\;\\ R_{\left(q_{w}^{i}\right)}\approx \left(I_{d_{3}}-{\left\lfloor \delta \theta_{w}^{i}\right\rfloor }_{\times }\right).R_{\left(q_{w}^{\hat{i}}\right)}. \end{array}$$

The vision algorithm yields the rotation from the camera frame into the vision frame $q_{v}^{c}$. We can model the error measurement as
(23)$${\tilde{z}}_{q}=z_{q}-{\hat{z}}_{q}=q_{i}^{c}⊗q_{w}^{i}⊗q_{v}^{w}⊗{\left(q_{i}^{c}⊗{\hat{q}}_{w}^{i}⊗{\hat{q}}_{v}^{w}\right)}^{-1}.$$

Finally, the measurements Jacobian *H* in $\tilde{z}=H.\tilde{x}$ is calculated based on the method in [33] and can be stacked together in the form
(24)$$\left[\begin{array}{c} {\tilde{z}}_{p}\\ {\tilde{z}}_{q} \end{array}\right]=\left[\begin{array}{c} H_{p}\\ \begin{array}{cccccc} 0_{3x6} & {\tilde{H}}_{q}^{wi} & 0_{3x10} & {\tilde{H}}_{q}^{ic} & 0_{3\times 3} & {\tilde{H}}_{q}^{vw} \end{array} \end{array}\right]\tilde{x}.$$
with the Jacobian matrices ${\tilde{H}}_{q}^{xy}$, known as the right Jacobian of SO(3), and are defined as:(25)$$\begin{array}{c} {\tilde{H}}_{q}^{xy}=J_{r}\left(\theta_{x}^{y}\right)=\lim_{\delta \theta \to 0}\frac{Log(Exp{\left(\theta \right)}^{*}⊗Exp\left(\theta +\delta \theta \right))}{\delta \theta },\;\\ J_{r}\left(\theta_{x}^{y}\right)=I_{d_{3}}-\left(\frac{1-cos∥\delta \theta ∥}{{∥\delta \theta ∥}^{2}}\right).{\left\lfloor \delta \theta_{x}^{y}\right\rfloor }_{\times }+\left(\frac{∥\delta \theta ∥-sin∥\delta \theta ∥}{{∥\delta \theta ∥}^{3}}\right).{\left\lfloor \delta \theta_{x}^{y}\right\rfloor }_{\times }^{2}. \end{array}$$

### 3.4. States Update

To update the framework for the current time step (k+1), we compute the innovation term *S*, Kalman gain *K*, and the states correction vector $\hat{\tilde{x}}$ defined as:(26)$$\begin{array}{ccc} S=HPH^{⊤}+R, & K=PH^{⊤}S^{-1}, & \hat{\tilde{x}}=K\tilde{z} \end{array}.$$

The error state covariance is updated as follows:(27)$$P_{k+1|k+1}=\left(I_{d_{28}}-KH\right)P_{k+1|k}{\left(I_{d_{28}}-KH\right)}^{⊤}+KRK^{⊤},$$
where $R_{\left[6x6\right]}=diag\left(R_{position},R_{orientation}\right)$ is the measurement noise covariance matrix.

The error quaternion is calculated by (Equation 3) to ensure its unit length, and then update the states vector: $X_{k+1}=X_{k}+\hat{\tilde{x}}$.

For the quaternions state update:(28)$${\hat{q}}_{k+1}=\frac{\left[\begin{array}{cccc} 1 & \frac{1}{2}\delta \theta_{k+1}^{1} & \frac{1}{2}\delta \theta_{k+1}^{2} & \frac{1}{2}\delta \theta_{k+1}^{3} \end{array}\right]⊗{\hat{q}}_{k}}{∥\left[\begin{array}{cccc} 1 & \frac{1}{2}\delta \theta_{k+1}^{1} & \frac{1}{2}\delta \theta_{k+1}^{2} & \frac{1}{2}\delta \theta_{k+1}^{3} \end{array}\right]⊗{\hat{q}}_{k}∥},$$
where $\delta \theta_{k+1}^{i}$ is the *i*th error state of this quaternion.

### 3.5. Reset Mode

The ES-EKF reset mode is performed by setting $\hat{\tilde{x}}\leftarrow 0$ and $P\leftarrow G.P.G^{⊤}$, where *G* is the Jacobian matrix defined by
(29)$$\begin{array}{c} G=diag(I_{d_{6}},J_{r_{wi}},I_{d_{10}},J_{r_{ic}},I_{d_{3}},J_{r_{vw}}),\;\\ J_{r_{xy}}=\frac{\partial \delta \theta_{x}^{y^{+}}}{\partial \delta \theta_{x}^{y}}=I_{d_{3}}-\frac{1}{2}{\left\lfloor \hat{\delta \theta_{x}^{y}}\right\rfloor }_{{}_{\times }}. \end{array}$$

## 4. Experiments

### 4.1. Setup

An extensive quantitative and qualitative evaluation is carried out to validate all the state estimation process aspects. This thorough performance analysis is run on the EuRoC benchmark [12] for an indoor system global positioning evaluation in low-speed flights and on the Fast Flight dataset [11] for outdoor experimentation at relatively high-speed flights. For a fair comparison, all the pipeline processing stages in both Algorithms 1 and 2 are performed on a 16 GB RAM laptop computer running 64-bit Ubuntu 20.04.3 LTS with AMD(R) Ryzen 7 4800 h × 16 cores 2.9 GHz processor and a Radeon RTX NV166 Renoir graphics card. In Table 1, we represent the quantitative insights of our experiment settings regarding the benchmarks statistical data and the sensors parameters in-detail.

The front-end of the pipeline, including both the data acquisition and pre-processing steps, is developed as a Python API that sends the optimization variables to the factor graph implemented in C++ using the Ceres solver [40] to achieve the lowest possible system latency before the state estimation process. The Sparse Normal Cholesky linear solver by the Ceres solver is employed to solve the least-squares convex optimization problem formulated in Equation (Equation 6) along with the Levenberg–Marquardt trust region strategy with the automatic differentiation tool for Jacobian calculations. The sparse Schur linear method is applied to utilize the Schur complement for a more robust and fast optimization process. The pipeline’s back-end for the state estimation process is developed entirely in MATLAB (https://github.com/AbanobSoliman/VIO_RGB_IMU (accessed on 30 October 2022)) and all the initialization parameters are given explicitly in Table 2.

The performance analysis is performed using the two trajectory evaluation metrics: root mean square error (RMSE) for the Fast Flight dataset compared to the GPS trajectory $p_{gps}$, and the RMS absolute trajectory error (ATE) for the EuRoC benchmark compared to the ground truth trajectory $T_{gt}$ provided with Vicon room sequences. The positional RMSE metric for the Fast Flight sequences is chosen because the ground truth GPS trajectories exist with unknown ground truth orientations. However, for EuRoC sequences, we select the RMS ATE metric for two reasons: 1. the Vicon system provides ground truth poses (positions and orientations); and 2. to ensure a fair comparison with the latest state-of-the-art methods based on the same error metric. The two trajectory evaluation metrics are formulated as follows:(30)$$\mathrm{RMSE}=\sqrt{\frac{1}{n}\sum_{i=1}^{n}{\left|\left|\hat{p}\left(i\right)-p_{gps}\left(i\right)\right|\right|}^{2}},\;\mathrm{ATE}=\sqrt{\frac{1}{n}\sum_{i=1}^{n}{\left||p(T_{gt}^{-1}\left(i\right).T_{rel}.\hat{T}\left(i\right))|\right|}^{2}}\;\left[m\right],$$
where $\hat{p}$ is the estimated translation vector of the $\hat{T}\in SE\left(3\right)$ trajectory; p(.) is the translation vector of the T∈SE(3) pose; and $T_{rel}$ is rigid-body transformation corresponding to the least-squares solution that maps the $\hat{T}$ trajectory onto the $T_{gt}$ trajectory calculated by optimization. We set it constant for all sequences that belong to the same benchmark.

### 4.2. The EuRoC MAV Benchmark

The two main characteristics of the EuRoC MAV sequences are the complex combined 6-DoF motions and the relatively low speeds compared to the Fast Flight sequences. These prominent characteristics allow an accurate evaluation of the ES-EKF marginally stable states, such as the velocity and the visual drift. In Table 3, we report the ATE values as an evaluation parameter for the trajectory estimation accuracy compared to the ground truth. Moreover, Table 3 shows an ablation study that investigates the contribution of the GPS sensor to the overall estimation accuracy, especially for the monocular vision-based optimization methods: ours (PGO) and the recent work of Cioffi et al. [30]. The selection of the six Vicon room sequences from the EuRoC benchmark is because a comparison with an alternative method such as [30] incorporating GPS signals simulated from the Vicon system readings better emphasizes the findings of this ablation study.

A prominent finding of this ablation study is that vision is the most significant type of sensor. In most sequences, the lowest ATE is obtained by fusing the camera trajectory from the vision KLT-based SfM algorithm to a gravity-aligned frame using the noisy simulated GPS data, and adding inertial measurements does not provide a measurable benefit in this case. However, adding the gyroscope measurements to the visual-GPS fusion has led to the least ATE achieved by our PGO model compared to all other discrete-time (DT) methods. Figure 6 and Figure 7 show our trajectory and velocity estimations after incorporating the accelerometer readings in the ES-EKF model, resulting in the lowest achievable errors that can compete with the continuous-time optimization model in [30].

### 4.3. The Fast Flight Dataset

The main observation, which is validated upon both the EuRoC and Fast Flight sequences (see Table 4 and Figure 8), is that for velocities less than 5 (m/s), the monocular loosely coupled ES-EKF can achieve considerably lower estimation errors concerning the other filter- or optimization-based methods. For velocities more than 5 (m/s), our proposed optimization-based initialization scores the lowest RMSE compared to all other methods in comparison in Table 4. On the contrary, the monocular ES-EKF scores the lowest RMSE, especially for velocities more than 10 (m/s), compared to the best-performing Kalman filter stereo model of the S-MSCKF.

Since the maximum achieved velocity of the EuRoC MAV is nearly 2.3 (m/s), the quantitative results in Table 3 further support this conclusion, where our ES-EKF scores the best performance compared to the other state-of-the-art methods. In-depth reasoning for this degraded performance at high speeds (more than 5 (m/s)) can be clarified based on the hardware characteristics of the MAV sensors’ properties, such as the data rate, latency, and noise effects at high speeds. Our optimization-based (PGO) initialization outperforms all other optimization- or filtering-based methods with high-rate visual-inertial sensors.

An insightful overview of the velocity profiles estimated by our ES-EKF is represented in Figure 9. The main conclusion is that the estimated velocity profile during the planar motion of the MAV in the X–Z plane optimally fits the upper and lower bounds of the top speed for each sequence. Towards an in-depth investigation to understand the high perturbations in the estimated velocity when approaching the maximum limit, we plot the velocity error states in the ES-EKF showing a high error at the instances when approaching top speeds due to the strong vibrations in the MAV structure affecting the IMU readings.

The high estimation accuracy of our ES-EKF model compared to GPS readings and the PGO optimization-based initialization process is further verified by the Y axis trajectory estimation in Figure 8. The maximum estimated altitude for all sequences by the ES-EKF is nearly 60 (m), whereas both the GPS readings and the initialization optimizer estimate a maximum altitude of nearly 100 (m). To physically validate which is a more accurate altitude estimation, we took snippets of the scene at a time instance in the exact halfway of all trajectories as shown in Figure 1. We can observe that the MAV is nearly on the same level as the roof of a commercial aircraft hangar, which is in the range of 30 (m) to 66 (m). This observation validates the high estimation accuracy of the altitude using our ES-EKF.

### 4.4. Real-Time Performance Analysis

The filter-based approaches are more advantageous for real-time onboard applications because they use the CPU more efficiently than the monocular and stereo optimization-based methods. Due to its computationally intensive front-end pipeline for both temporal and stereo matching, OKVIS uses more CPU than VINS-Mono. Additionally, OKVIS’s back-end operates at a speed that is much faster than the set 10 (Hz) rate of VINS-Mono. Approximately 90% of the work in our back-end, ES-EKF, is brought on by the front-end, which includes ORB feature detection, KLT-based tracking, and matching. At 200 (Hz), the filter uses approximately 10% of a core. Our suggested technique offers the maximum estimation frequency, which provides the optimal balance between the precision and computing cost.

Figure 10 contrasts how much CPU time various VIO solutions used on the EuRoC benchmark and the Fast Flight dataset. Since V2-03 has considerable scale drift with S-IEKF and S-UKF-LG techniques and hence has significantly worse accuracy when compared to other methods, the CPU consumption of V2-03 is excluded from the comparison. According to the testing, the ES-EKF achieves the lowest CPU consumption while retaining a similar level of accuracy in comparison with other methods. We notice that the proposed method puts more computing work into the image processing front-end than the tests using the EuRoC dataset. Higher imaging frequency and resolution are one explanation, while Fast Flight results in a shorter feature lifetime, necessitating frequent new feature identification, is another reason.

## 5. Conclusions

Our work aimed to provide an accurate and computationally inexpensive localization solution during MAVs’ long-term navigation in large-scale environments. We represented a loosely coupled IMU/GPS camera fusion framework with pose failure detection methodology toward this goal. Moreover, we proposed a novel decoupled optimization- and filtering-based sensor fusion technique that achieves a high estimation accuracy and minimum system complexity compared to the other methods in the literature. We used real-world indoor and outdoor settings for the MAV localization studies to validate and test the findings of our proposed method.

The vision-based black-box pose estimation accuracy is first examined in a controlled laboratory Vicon room of the EuRoC benchmark. The outcomes confirmed the system’s reliance on monocular vision. The experiments on EuRoC and Fast Flight sequences have shown remarkable accuracy in the trajectory estimation studies. We also evaluated the proposed scheme in terms of computational complexity, measured by CPU usage, where our monocular-vision optimization/filtering solution outperformed all the competing techniques.

This conclusion enforces our work’s contributions to a reliable (fast and accurate) sensor fusion solution for challenging and large-scale environments. From a future perspective, it will be necessary to comprise situations where GPS sensor constraints, such as the multipath effects on the optimizer. Finally, further generalizing the optimization problem will be necessary to extend the algorithm’s pose estimation capability to include multiple vision sensors (stereo RGB, for instance).

## Figures and Tables

**Figure 1 sensors-23-00516-f001:**
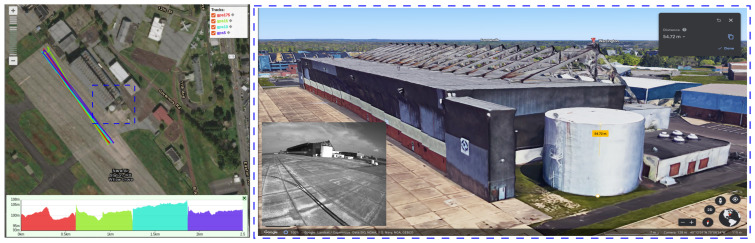
An example for the on-map GPS readings of the large-scale environment of the Fast Flight dataset [11] sequences: gps175, gps15, gps10, and gps5. The sequence number denotes the maximum flight velocity of each sequence: 17.5, 15, 10, and 5 (m/s), respectively. The color bar (bottom) denotes the map scale in (km) on the x axis and the altitude of each sequence in (m) on the y axis. In the blue dotted box: Comparing the maximum MAV’s altitude at instance before the descent stage to the height of an aircraft hangar. The estimated airport asset height is 54.72 (m), corresponding to the maximum MAV altitude. Images are courtesy of Google Earth.

**Figure 2 sensors-23-00516-f002:**
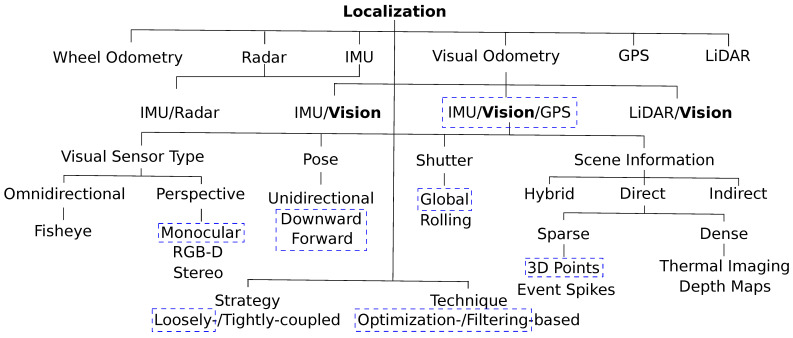
Visual odometry is generally categorized together with self-contained and global localization methods.

**Figure 3 sensors-23-00516-f003:**
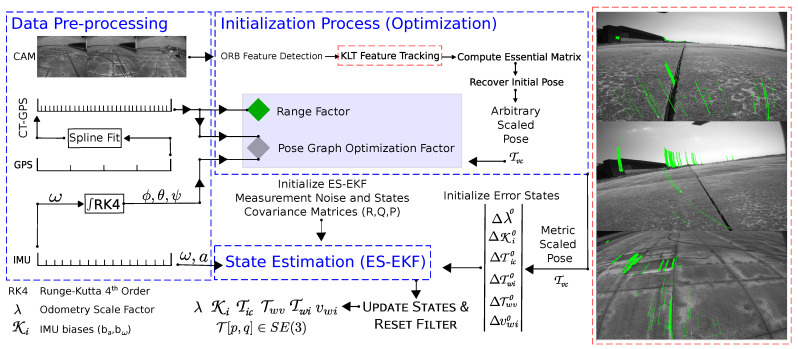
Overview of our proposed entire system architecture.

**Figure 4 sensors-23-00516-f004:**
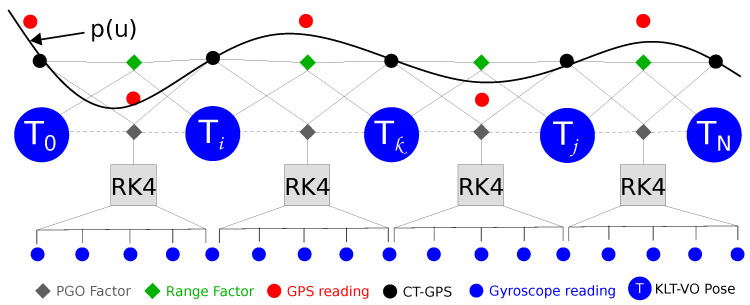
Initialization factor graph. p(u) is the CT-GPS trajectory generated at high frequency. RK4 is the Runge–Kutta 4th order gyroscope integration scheme. Dotted lines denote the error term $\left({\hat{T}}_{i}^{-1}{\hat{T}}_{j}\right)$ in Equation (Equation 7) between any two KLT-VO poses.

**Figure 5 sensors-23-00516-f005:**
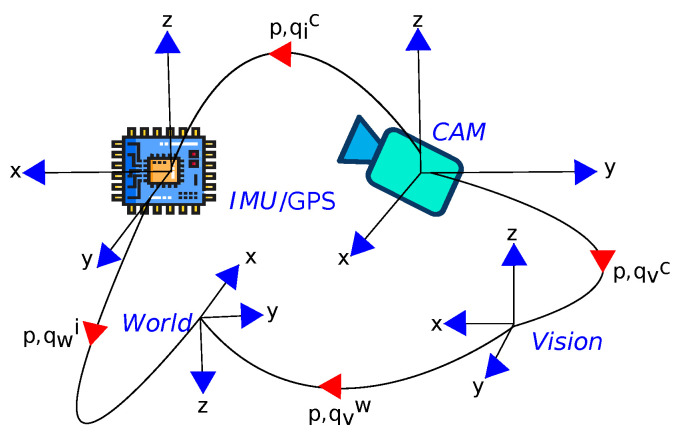
The frames of reference annotations.

**Figure 6 sensors-23-00516-f006:**
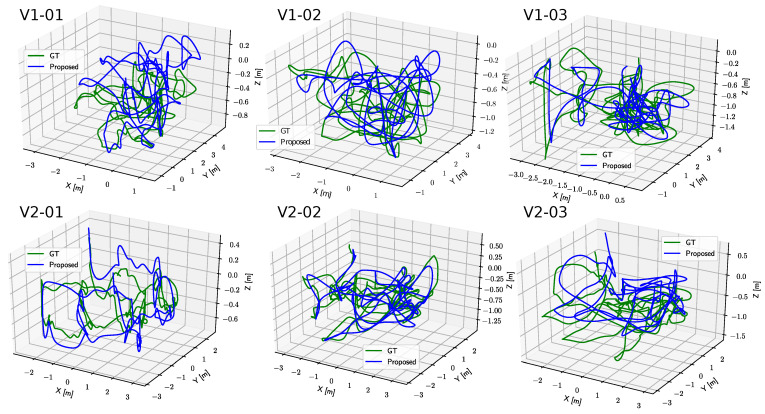
EuRoC 3D trajectory estimation compared to the ground truth.

**Figure 7 sensors-23-00516-f007:**
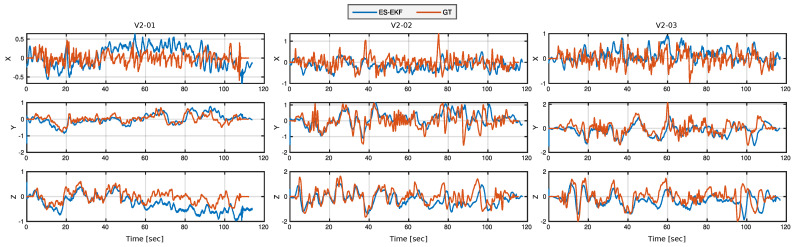
Estimated velocity profile validation with the ground truth. Comparison of sample sequences from EuRoC benchmark.

**Figure 8 sensors-23-00516-f008:**
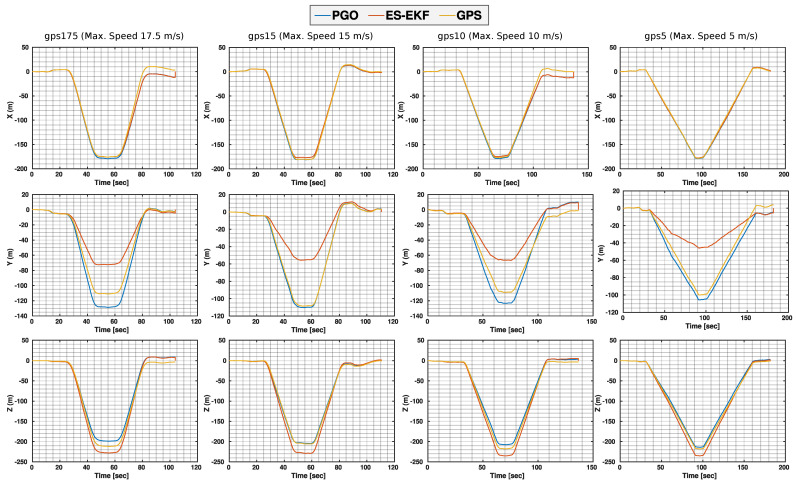
Fast Flight (X (**top**)−Y (**middle**)−Z (**bottom**)) trajectory estimation compared to the GPS readings.

**Figure 9 sensors-23-00516-f009:**
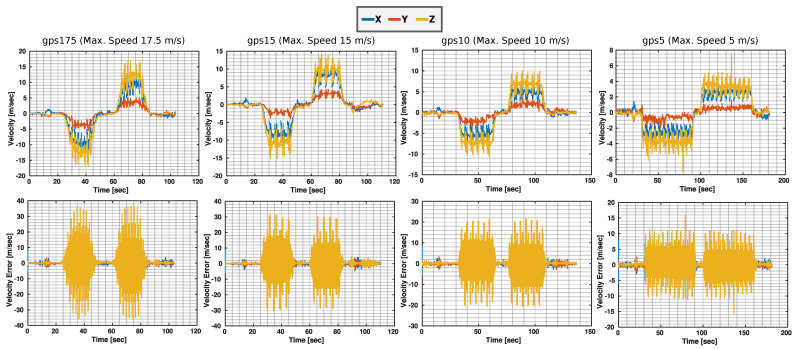
(**Top**): Fast Flight velocity profile validation with the top speed of each sequence. (**Bottom**): velocity error states in the ES−EKF.

**Figure 10 sensors-23-00516-f010:**
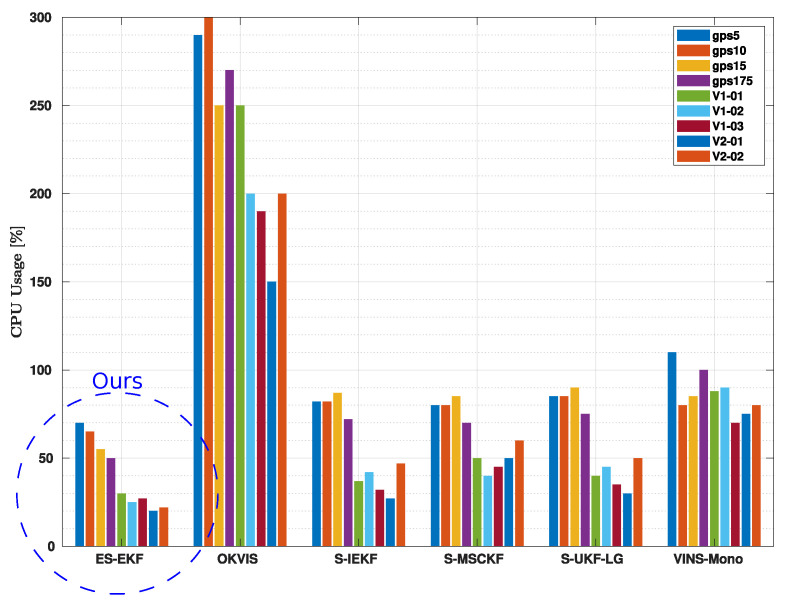
CPU usage as a real-time performance analysis indicator.

**Table 1 sensors-23-00516-t001:** Insights into our experiments’ statistical information and sensor settings.

Parameter		EuRoC Benchmark [12]				Fast Flight Dataset [11]			
Stats	Total processed sequences	6 (Vicon room)				4 (airport runway)			
	Total sequences duration	11.6111 min				8.8867 min			
	Total sequences length	411.5425 m				2539.0599 ^1^ m			
	Maximum speed	2.3 (m/s)				17.5 (m/s)			
Camera	Total processed frames	13,736				21,312			
	Frame resolution	752 × 480 pixels				960 × 800 pixels			
	Intrinsics $\left(f_{x},f_{y},c_{x},c_{y}\right)$	458.65	457.30	367.22	248.38	606.58	606.73	474.93	402.28
	Distortion $\left(k_{1},k_{2},p_{1},p_{2}\right)$	−0.2834	0.0739	0.0001	0.000018	−0.0147	−0.0058	0.0072	−0.0046
	Camera-IMU $p_{i}^{c}$(x,y,z,1) (m)	−0.0216	−0.0647	0.0098	1.0000	0.1058	−0.0177	−0.0089	1.0000
	Camera-IMU $q_{i}^{c}$(x,y,z,w) [-]	−0.0077	0.0105	0.7018	0.7123	−1.0000	0.0042	−0.0039	0.0015
	Frame rate	20 (Hz)				40 (Hz)			
IMU	Gyroscope noise density ($\sigma_{n_{\omega }}$)	$1.6968\times 10^{-4}$ [rad/s/$\sqrt{\mathrm{Hz}}$]				$6.1087\times 10^{-5}$ [rad/s/$\sqrt{\mathrm{Hz}}$]			
	Gyroscope random walk ($\sigma_{n_{b_{\omega }}}$)	$1.9393\times 10^{-5}$ [rad/s${}^{2}$/$\sqrt{\mathrm{Hz}}$]				$9.1548\times 10^{-5}$ [rad/s${}^{2}$/$\sqrt{\mathrm{Hz}}$]			
	Accelerometer noise density ($\sigma_{n_{a}}$)	$2.0000\times 10^{-3}$ [m/s${}^{2}$/$\sqrt{\mathrm{Hz}}$]				$1.3734\times 10^{-3}$ [m/s${}^{2}$/$\sqrt{\mathrm{Hz}}$]			
	Accelerometer random walk ($\sigma_{n_{b_{a}}}$)	$3.0000\times 10^{-3}$ [m/s${}^{3}$/$\sqrt{\mathrm{Hz}}$]				$2.7468\times 10^{-3}$ [m/s${}^{3}$/$\sqrt{\mathrm{Hz}}$]			
	Data rate (1/Δt)	200 (Hz)				200 (Hz)			
GPS	Type/operation	Indoors/Vicon system				Outdoors/satellite Triangulation			
	Readings	X (m), Y (m), Z (m)				Long. (deg), Lat. (deg), Alt. (m)			
	Data rate	1 (Hz) (down-sampled)				5 (Hz)			

^1^ Denotes the exact value of the total trajectories lengths for all of the sequences of Fast Flight dataset shown on the x axis of Figure 1 (≈2.5 (km)).

**Table 2 sensors-23-00516-t002:** The ES-EKF initialization parameters for both the EuRoC and Fast Flight sequences.

Parameter Initialization	EuRoC Benchmark [12]	Fast Flight Dataset [11]
28-element error state vector ($\hat{\tilde{x}}$)	$0_{28\times 1}$	$0_{28\times 1}$
31-element state vector ^1^ (X)	${\left(0_{3\times 1}\;0_{3\times 1}\;{\bar{q}}^{⊤}\;0_{3\times 1}\;0_{3\times 1}\;1\;{p_{i}^{c}}^{⊤}\;{q_{i}^{c}}^{⊤}\;0_{3\times 1}\;{\bar{q}}^{⊤}\right)}^{⊤}$	
States propagation covariance (*P*)	$10^{-7}\times I_{d_{28}}$	$10^{-12}\times I_{d_{28}}$
CT process noise covariance ^2^ ($Q_{c}$)	$diag(d\sigma_{n_{a}}^{2}.I_{d_{3}},d\sigma_{n_{b_{a}}}^{2}.I_{d_{3}},d\sigma_{n_{\omega }}^{2}.I_{d_{3}},d\sigma_{n_{b_{\omega }}}^{2}.I_{d_{3}})$	
Measurement noise covariance (R)	$diag(0.01,0.01,0.03,10^{-4},10^{-4},10^{-4})$	

^1^$\bar{q}$ denotes the unity quaternion [0,0,0,1]. ^2^ IMU noise density values for each dataset are from Table 1 and discretized using Equation (Equation 11).

**Table 3 sensors-23-00516-t003:** Ablation study on the contribution of the GPS sensor on the system accuracy. The latest state-of-the-art (monocular/stereo) VI-SLAM systems are compared to our proposed trajectory initialization (PGO factors) and ES-EKF state estimation methods. **Bold** denotes the most accurate.

Method		EuRoC Benchmark [12] (RMS ATE [m])						Avg.
		V1-01	V1-02	V1-03	V2-01	V2-02	V2-03	
Mono-VI	OKVIS [23]	0.090	0.200	0.240	0.130	0.160	0.290	0.185
	ROVIO [20]	0.100	0.100	0.140	0.120	0.140	0.140	0.123
	VINS-Mono [22]	0.047	0.066	0.180	0.056	0.090	0.244	0.114
	OpenVINS [41]	0.056	0.072	0.069	0.098	0.061	0.286	0.107
	CodeVIO ^1^ [42]	0.054	0.071	0.068	0.097	0.061	0.275	0.104
	Cioffi et al. ^2^ [16]	0.034	0.035	0.042	0.026	0.033	0.057	0.038
Stereo-VI	VINS-Fusion [13]	0.076	0.069	0.114	0.066	0.091	0.096	0.085
	BASALT [25]	0.040	0.020	0.030	0.030	0.020	0.050	0.032
	Kimera [43]	0.050	0.110	0.120	0.070	0.100	0.190	0.107
	ORB-SLAM3 [24]	0.038	0.014	0.024	0.032	0.014	0.024	0.024
Mono-(V/I/G) ^3^	CT (V+I+G) [30]	0.024	0.014	**0.011**	0.012	0.010	**0.010**	0.014
	CT (V+G) [30]	0.011	0.013	0.012	0.009	**0.008**	0.012	**0.011**
	CT (I+G) [30]	0.062	0.102	0.117	0.112	0.164	0.363	0.153
	DT (V+I+G) [30]	0.016	0.024	0.018	0.009	0.018	0.033	0.020
	DT (V+G) [30]	0.010	0.025	0.024	0.010	0.012	0.029	0.018
	DT (I+G) [30]	0.139	0.137	0.138	0.138	0.138	0.139	0.138
	**Ours (PGO)**	**0.008**	0.017 ^4^	0.023 ^4^	**0.008**	0.022	0.025 ^4^	0.017
	**Ours (ES-EKF)**	0.009	**0.012**	**0.011**	0.010	0.011	**0.010**	**0.011**

^1^ Denotes the only learning-based baseline in the table and incorporates point clouds using LiDAR. ^2^ Denotes values from the original work with four GPS readings connected to each optimization state. ^3^ V,I,G: Vision, IMU, and GPS (generated from the Vicon system readings). ^4^ Denotes KLT-VO tracks features in 5 consecutive frames instead of 10 due to the rapid movement of the MAV.

**Table 4 sensors-23-00516-t004:** Ablation study on the effect of the high MAV speed on the accuracy of the filtering approaches compared to optimization approaches. The first sub-section compares monocular (VINS-Mono and Ours) to stereo (OKVIS) optimization-based VI systems. The second sub-section compares stereo filtering-based approaches to our proposed method. **Bold** denotes the most accurate in each sub-section.

Method	Fast Flight [11] (RMSE (m))				Avg.
	gps5	gps10	gps15	gps175	
OKVIS [23]	3.224	4.987	3.985	4.535	4.183
VINS-Mono [22]	5.542	8.753	2.875	3.452	5.156
**Ours (PGO)**	**0.417**	**0.759**	**0.180**	**0.927**	**0.571**
S-MSCKF [11]	4.985	2.751	**4.752**	**7.852**	**5.085**
S-UKF-LG [19]	4.875	2.589	5.128	7.865	5.114
S-IEKF [19]	4.986	**2.544**	5.124	8.152	5.201
**Ours (ES-EKF)**	**4.751**	7.924	7.221	9.488	7.346

## Data Availability

The EuRoC MAV benchmark is publicly available online at (https://projects.asl.ethz.ch/datasets/doku.php?id=kmavvisualinertialdatasets (accessed on 1 October 2022)). The Fast Flight dataset is publicly available online at (https://github.com/KumarRobotics/msckf_vio/wiki/Dataset (accessed on 3 October 2022)).

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
