# Peer review of "MAV Localization in Large-Scale Environments: A Decoupled Optimization/Filtering Approach"

_sensors, 2023, doi:10.3390/s23010516_

Round 1

Reviewer 1 Report

A novel Micro Aerial Vehicles localization method is proposed. 

Figure 1 with the on-map GPS readings example must be cited in section 1.

It is necessary to add a brief discussion of the overview of the current localization methods in Fig. 2. Furthermore, authors must point out why the methodologies in blue dashed rectangles in Fig. 2 inspired their research.

A brief discussion on the future perspectives of the research should be added in the concluding section.

A more accurate description of the comparative results shown in Tables 3 and 4 is needed. For example, in Tab. 4, relative to stereo filtering-based methods, on average the most accurate results seem to be those obtained by S-MSCKF, while ES- EKF has a very high average RMSE. Authors need to add a discussion on these points.

Reviewer 2 Report

In the article " MAVs Localization in Large-scale Environments: A decoupled Optimization/Filtering Approach", the authors present a linear optimal state estimation approach for the MAV to avoid complicated and high-latency calculations and an immediate metric-scale recovery paradigm that uses low-rate noisy GPS measurements when available. Autors proposed strategy the vision sensor can quickly bootstrap a pose that has been arbitrarily scaled and recover from various drifts that affect vision-based algorithms. They considered the camera as a "black-box" pose estimator thanks to our proposed optimization/filtering-based methodology. This maintains the sensor fusion algorithm’s computational complexity and makes it suitable for MAV’s long-term operations in expansive areas. Due to the limited global tracking and localization data from the GPS sensors, our proposal on MAV’s localization solution considers the sensor measurement uncertainty constraints in such circumstances. Extensive quantitative and qualitative analyses utilizing real-world and large-scale MAV sequences, demonstrates the higher performance of our technique in comparison to the most recent state-of-the-art algorithms in terms of trajectory estimation accuracy and system latency.

In the introduction, the authors provided an overview of the available localization methods and reviewed these methods and highlighted the need to develop a state estimation approach that should consistently have low computational complexity and be based on abnormal sensor readings. The Aurors also proposed their contribution, i.e.

- In case of state estimator initialization failure, proposed a unique instant bootstrapping technique based on continuous-time manifold optimization via Pose Graph Optimization (PGO) and Range factors, which depends on low-rate GPS signals.

- A closed-form estimation method without non-linear optimization during IMU/CAM fusion produces a reduced system latency with a constant CPU computing complexity. The mathematical modeling of a linear ES-EKF with a precise and quick gyroscope integration strategy accounts for the simplicity of proposed localization solution.

- The EuRoC benchmark [12], for MAV localization assessment in indoor environments, and the Fast Flight dataset [11], for large-scale outdoor environments, are two real-world publicly available benchmarks on which IMU/GPS-CAM fusion system

In chapters 2 authors described the related works. As a difference to the methods used so far, the authors indicated a simple approach to spline fitting for GPS readings at the stage of data preprocessing.

In Chapter 3, the authors described the architecture of the system. The Structure from Motion (SfM) incremental algorithm is used for the acquired image frames, the purpose of which is to recreate the camera position and the three-dimensional structure of the scene, based on a five-point algorithm. ORB features are detected and the highest quality points are tracked between 10 consecutive frames using the KLT method. To solve the problem of the arbitrary scale of only the camera trajectory, we apply B-spline3 onmanifold cumulative interpolation to synthesize a very smooth continuous time (CT) trajectory in R3 from noisy low-speed GPS readings.

In Chapter 4, the authors presented the results of the experiments. Real-time performance analysis is noteworthy. The authors showed that filter-based churns are better for real-time on-board applications because they use the processor more efficiently than methods based on monocular and stereo optimization. Because of his

computationally intensive front-end for both temporal and stereo matching, OKVIS uses more CPU than VINS-Mono. Additionally, the OKVIS back-end runs at a speed much greater than the set frequency of 10 [Hz] VINS-Mono. About 90% of the work in the ES-EKF backend authors solution is done by the front-end, which includes ORB function discovery, KLT-based tracking and matching. At 200 [Hz], the filter itself consumes about 10% of the core. The proposed technique offers the maximum frequency of estimation, which ensures an optimal balance between the precision and the cost of calculations. According

testing ES-EKF achieves the lowest CPU usage while maintaining a similar level of accuracy compared to other methods. The authors noted that the proposed method imposes more computational work on the front-end of image processing than tests using the EuRoC dataset. Higher imaging frequency and resolution is one of the explanations, while Fast Flight results in shorter feature lifetime, necessitating frequent identification of new features.

In conclusion, the authors indicated that their work was aimed at providing an accurate and computationally inexpensive positioning solution for long-term MAV navigation in large-scale environments. They presented a loosely coupled IMU / GPS-Kamera fusion structure with failure detection methodology to achieve this goal. Moreover, they proposed a novel sensor connection technique based on separate optimization and filtering, which ensures high accuracy of estimation and minimal system complexity compared to other methods described in the literature. They used actual internal and external settings for MAV location studies to verify and test the results of the proposed method. The accuracy of the vision black box pose estimation is first tested in a controlled Vicon laboratory under the EuRoC benchmark. The results confirmed the dependence of the system on monocular vision. Experiments on the EuRoC and Fast Flight sequences have shown remarkable accuracy in trajectory estimation studies. They also assessed the proposed scheme for computational complexity, as measured by CPU utilization, where their monocular vision optimization / filtering solution outperformed all competing techniques. They also set out to further develop the technique in order to achieve a reliable (fast and accurate) sensor, i.e. a fusion solution for demanding and large-scale environments.

The article is well prepared. The topics discussed were supported by a broad literature review, and the proposed method was tested both in a laboratory and in a real environment. I recommend publishing the article.

Reviewer 3 Report

The contribution of this work is  interesting.  The title is descriptive. The abstract clearly indicates the scope. The paper is well organised and logically written, nevertheless, the English language of  the contribution should be improved by a native speaker.

Appropriate research goals are chosen in this contribution, which shows that the authors have a high level of understanding of current research within the field. The presentation of the results in terms of the research objectives has been  made, nevertheless there should be a deeper clarification.

The authors have been able to draw logical conclusions from the results. The experimental results show that the adaptive Kalman filter improves the achieved score and in total the feasibility of the proposed vision based interface, which has reasonable performance and response time comparing with the standard game pad.

The quality of pictures and figures is good.

The contribution needs to be improved after considering the points indicated here below:

. In (6) an optimization problem is formulated.  Please explain the used method to calculate the solution better. Is this optimization problem an convex one?

- In (30)  two figure of merit are considered. Please explain more in depth the motivation of this choice. In fact, it is known that the quality of the results depends on this choice.

Minor aspects:

The results are very well descripted, but please imporve the quality of the figure if it is possible. In fact, the ledenda together with the labels are too small to be read! 

These following papers can be useful for readers in the context of the application in the field of Kalman Filter. Some of them published in Sensors of MDPI publisher.

Dai, J.; Liu, S.; Hao, X.; Ren, Z.; Yang, X. UAV Localization Algorithm Based on Factor Graph Optimization in Complex Scenes. Sensors 2022, 22. https://doi.org/10.3390/s22155862.

An extended Kalman filter as an observer in a control structure for health monitoring of a metal–polymer hybrid soft actuator, M Schimmack et al. 2018, IEEE/ASME Transactions on Mechatronics 23 (3), 1477-1487.

Ren, G.; Yu, Y.; Liu, H.; Stathaki, T. Dynamic Knowledge Distillation with Noise Elimination for RGB-D Salient Object Detection. Sensors 2022, https://doi.org/10.3390/s22166188.

Cioffi, G.; Cieslewski, T.; Scaramuzza, D. Continuous-Time Vs. Discrete-Time Vision-Based SLAM: A Comparative Study. IEEE 507 Robotics Autom. Lett. 2022, 7, 2399–2406. https://doi.org/10.1109/LRA.2022.3143303.

A switching kalman filter for sensorless control of a hybrid hydraulic piezo actuator using mpc for camless internal combustion engines, P.  Mercorelli, 2012 IEEE International Conference on Control Applications, 980-985

Round 2

Reviewer 1 Report

Authors revised this manuscript taking into account all my suggestions. I consider this paper publishable in the current form.